



# Storm-driven across-shelf oceanic flows into coastal waters

Sam Jones[a*], Mark Inall[ab], Marie Porter[a], Jennifer A. Graham[c], Finlo Cottier[ad]

[a] Scottish Association for Marine Science, Oban, Argyll, PA371QA, UK
[b] Also with: University of Edinburgh, School of Geosciences, Edinburgh, EH93FE, UK
[c] Centre for Environment, Fisheries and Aquaculture Science, Pakefield Road, Lowestoft, NR330HT,UK
[d] Also with: Faculty of Biosciences, Fisheries and Economics, UiT The Arctic University of Norway, Tromsø,
Norway

*Correspondence to: Sam Jones (Sam.Jones@sams.ac.uk)

**Abstract.** The North Atlantic Ocean and Northwest European shelf experience intense low-pressure systems
during the winter months. The effect of strong winds on shelf circulation and water properties is poorly
understood as observations during these episodes are rare, and key flow pathways have been poorly resolved by
models up to now. We compare the behaviour of a cross-shelf current in a quiescent period in late summer,
with the same current sampled during a stormy period in mid-winter, using drogued drifters. Concurrently,
high-resolution time-series of current speed and salinity from a coastal mooring are analysed. A Lagrangian
analysis of modelled particle tracks is used to supplement the observations. Current speeds at 70 m during the
summer transit are 10-20 cm s$^{-1}$, whereas on-shelf flow reaches 60 cm s$^{-1}$ during the winter storm. The onset of
high across-shelf flow is identified in the coastal mooring time-series, both as an increase in coastal current
speed and as an abrupt increase in salinity from 34.50 to 34.85, which lags the current by 8 days. We interpret
this as the wind-driven advection of outer-shelf (near-oceanic) water towards the coastline, which represents a
significant change from the coastal water pathways which typically feed the inner shelf. The modelled particle
analysis supports this interpretation: particles which terminate in coastal waters are recruited locally during the
late summer, but recruitment switches to the outer shelf during the winter storm. We estimate that during
intense storm periods, on-shelf transport may be up to 0.48 Sv, but that this is near the upper limit of transport
based on the multi-year time series of coastal current and salinity. The likelihood of storms capable of
producing these effects is much higher during NAO-positive winters.

**1. Introduction**

The shelf seas and abyssal ocean are often treated as separate systems as they exhibit very different behaviours
despite the lack of a physical barrier between the two. The shallow continental shelves are responsible for
roughly 25 % of global primary production (Simpson and Sharples, 2012) and play host to the vast majority of
human-ocean interactions. The exchange of water between the ocean and shelf seas is still an evolving field of
study, as is our understanding of how oceanic and shelf edge processes play out at the coast (Brooks and
Townsend, 1989; Holt et al., 2009; Huthnance et al., 2009; Münchow and Garvine, 1993). In this paper we
investigate the behaviour of a newly characterised across-shelf current, the Atlantic Inflow Current (AIC, Porter
et al., (2018)) and the role it plays in transporting oceanic water across the Northwest European shelf.

The Northwest European shelf is bounded to the west by the Northeastern Atlantic and Rockall trough basins
(Fig. 1a). A topographically steered slope current flows along the shelf edge between Biscay and Norway,





becoming increasingly consistent in flow speed and direction north of the Celtic Sea. The stability and persistence of the slope current, particularly north of 55 °N favours along-slope (poleward) transport, and inhibits ocean-shelf exchange (Huthnance et al., 2009; White and Bowyer, 1997). Despite the reduced cross-shelf flow associated with the slope current, the presence of relatively undiluted oceanic water is detectable

many tens of km on-shelf (Inall et al., 2009; Jones et al., 2018; Jones, 2016) at several persistent locations along the shelf. These regions of oceanic incursion are important for the delivery of production-limiting oceanic nutrients to coastal seas (Painter et al., 2016) and by extension, the level of primary production and prospects of local fisheries (Gowen et al., 1998; Miller, 2013; Proctor et al., 2003). The waters of the Northwest European shelf exhibit a general clockwise circulation around the UK, and if oceanic water remains on the shelf it will

ultimately be mixed into this coastal current system (Hill, 1983; Simpson and Hill, 1986). Thus, these oceanic incursions have the potential to influence not only local coastal seas, but also those downstream, for example the North Sea. This on-shelf transport implies a balancing off-shelf flow of water, and the properties of the North Atlantic Current are gradually modified as it progresses polewards through its interaction with shelf seas (Holliday et al., 2000; Reid et al., 2001)

While schematics of shelf circulation typically infer a permanent residency of oceanic water on regions of the shelf (Ellett and Edwards, 1983; Ellett and MacDougal, 1985; Inall et al., 2009; Simpson et al., 1979), in reality these intrusions appear to be sporadic both in duration and geographic extent. Evidence for the spatial mobility of fronts and water masses on the Malin shelf is provided by satellite observations, as well as in-situ cruise and mooring data (Ellett and Edwards, 1983; Inall et al., 2009; Jones et al., 2018; Jones, 2016; Porter et al., 2018).

The variable occupation of the Malin shelf by oceanic water means that coastal water properties in some regions exhibit high temporal variability (Jones et al., 2018). We seek to characterise the causes and nature of these oceanic intrusions.

Our investigation builds on findings from a drifter release on the Malin shelf during the FASTNEt shelf edge observation campaign (Porter et al., (2018) , Fig. 1b). Drogued drifters released into the slope current in July

2013 moved on-shelf downstream of a canyon system at 55.5 °N and travelled towards the coast in a coherent current which the authors named the AIC. While on-shelf flow is captured by models of the region (Aleynik et al., 2016; Graham et al., 2018b; Holt et al., 2009; O'Dea et al., 2012; Xing and Davies, 2001; Young and Holt, 2007), this study provided the first evidence of a narrow, jet-like current crossing $f/h$ contours and transporting oceanic water onto the adjacent shelf.

We also utilise salinity and current observations from a fixed mooring in Tiree Passage off the west coast of Scotland. The Tiree Passage Mooring (TPM) tracks the highly variable mix of coastal water, freshwater runoff and oceanic water that flows through the Inner Hebrides, collectively referred to as the Scottish Coastal Current (SCC). The SCC originates in the baroclinically-driven outflow from the Irish Sea (Hill, 1987; Hill and Simpson, 1988; Jones, 2016), and receives contributions from rivers and sea-lochs, causing it to become less

saline and increase in volume as it progresses northward. Residual currents at the TPM are typically poleward with an average speed of 10 cm s$^{-1}$ though with much variability at timescales between a few hours and several weeks (Inall et al., 2009). The salinity measured by the TPM is highly variable on inter-annual timescales, but sometimes also features abrupt changes over a few hours or days (Fig. 2). Salinity has proved to be a sensitive tracer of water masses on the Northwest European shelf and was found by Jones et al., (2018) to track the



relative positions and concentrations of oceanic (S > 35.2) and coastal (S < 34.9) water masses on the shelf. The
main mode of variability in salinity was found by the authors to be due to a wind control mechanism on the
origin of the SCC, such that sustained easterly winds enhanced outflow from the Irish Sea, whereas sustained
westerly winds retarded it. This work built on earlier observations of wind-control on the North Channel of the
Irish Sea (Bowden and Hughes, 1961; Brown and Gmitrowicz, 1995), observations of temperature and current

speed in Tiree Passage (Inall et al., 2009) and model studies indicating that wind control of current pathways
may extend to the wider Northwest European shelf (Davies and Xing, 2003; Xing and Davies, 2001).

Once or twice during most winters a brief pulse of very high salinity water is observed at the TPM and these
high salinity pulses (HSPs) are often associated with storm events (Jones et al., 2018). An HSP constitutes a
brief but significant change from usual circulation patterns because nearly undiluted oceanic water is observed

at a coastline normally buffered from the Atlantic by the SCC. HSPs may be an important mediator of winter
coastal water properties as oceanic water is a source of both heat and nutrients to the shelf seas (Painter et al.,
2016; Porter et al., 2018; Siemering et al., 2016). They may have other impacts at the coast, for instance the
import of organisms typically restricted to waters further offshore.

Understanding of the drivers of these events is scant: while HSPs are associated with storm activity, salinity at

the TPM does not correlate simply with wind forcing on the shelf (Jones et al., 2018). Also, it is not known
whether accepted transport pathways from the outer shelf hold true when wind and wave action mask the
weaker baroclinic flows which prevail during quiescent periods. Observations of HSPs are few as they typically
occur in mid-winter when most satellite sensors are obscured by cloud. In addition, they are associated with
stormy periods in a region notorious for rough seas so are generally not captured by oceanographic cruises

which cannot sample in such conditions.

This study capitalises on the fortuitous recirculation of two drifters out of the 30 released in July 2013 during the
FASTNEt programme (Porter et al., 2018). While most drifters had exited the Malin shelf by October 2013,
two drifters drogued at 70 m were captured by an eddy in the Rockall Trough shortly after release, and only
crossed onto the Malin shelf in December 2013. This transit coincided with an HSP being measured by the

TPM so has the potential to explain the origin and nature of these phenomena. We therefore examine the
Lagrangian properties of the December drifter tracks in comparison with the fixed time series at the TPM. In
addition, we compare the shelf conditions during this winter shelf transit with the late summer conditions
sampled by the first cluster of drifters. This suite of observations is complemented by a modelled particle
tracking experiment.

**2. Methods**

**2.1. Drogued drifter release**

On 17th July 2013, 30 satellite tracked GPS drifters were released from the RRS James Cook on the 600 m
contour at 55.2 °N. The drifters were MetOcean SVP (Surface Velocity Program) drifting buoys fitted with a
holey-sock drogue (Sybrandy et al., 2009). Within this release, 15 of the drifters were drogued to track water at

15 m depth in the mixed layer, and 15 were drogued at 70 m to track water at the bottom of the seasonal
pycnocline. The drifters were fitted with a strain gauge which enabled checks for grounding, snagging and
drogue loss to be performed. Instances of unusual drifter displacement due to vessel interference were also





identified and subsequent observations rejected. For further information on quality control of drifter data, see
Porter et al., (2018).

To exclude high frequency motions such as tides and inertial oscillations from the drifter tracks, the data were
        filtered with a 10th order zero phase Butterworth low pass filter with a cut off at 2 cpd. Gaps in positional data
        were linearly interpolated with a maximum gap size of 20 hours. Drifter velocities were calculated using
        displacements from the filtered location data.

### 2.2. Tiree Passage Mooring (TPM)

The TPM is situated in northern Tiree Passage at 56.6 °N, 6.4 °W in 45 m deep water. Hourly current and
        temperature data were collected at the mooring using Anderaa current meters between 1981 and 2014 at a
        nominal depth of 20 m. Reliable salinity measurements commenced in 2002 with the addition of a Seabird
        Microcat to the standard array. The mooring was serviced at 3-5 monthly intervals by staff at the Scottish
        Association for Marine Science (SAMS), Oban though many gaps exist in the time series due to the challenging
conditions often found in Tiree Passage. CTD casts were conducted at the beginning and end of most mooring
        deployments to aid calibration of the fixed instruments. For more information on calibration and quality control
        of the TPM dataset, see Jones et al., (2018).

### 2.3. ECMWF forecast reanalysis data

        Daily 10 m wind and sea level pressure data from the ERA-Interim 0.75° x 0.75° product were obtained from
the European Centre for Medium-Range Weather Forecasts (ECMWF, Dee et al. (2011)). The ERA-Interim
        product was chosen over the more recent ERA5 release as the former was used to force the model used in this
        study.

### 2.4. AMM15 model data

        This study uses model output from AMM15 (Atlantic Margin Model, 1.5 km resolution), developed and
validated by Graham et al. (2018a) and Graham et al. (2018b) through the UK Joint Weather and Climate
        Research Programme. This incarnation of the model builds on AMM7 (7 km resolution) which has been
        utilised and validated by numerous studies (O'Dea et al., 2017, 2012). Both models are based on NEMO
        (Nucleus for European Modelling of the Ocean) architecture.

        AMM15 bathymetry is derived from EMODnet (EMODnet Portal, September 2015 release), and it uses a z* - σ
coordinate system (Siddorn and Furner, 2013) with 51 vertical levels. This hybrid system utilises terrain-
        following coordinates fitted to a smoothed envelope bathymetry. Vertical turbulent viscosity and diffusivity is
        calculated used the generic length scale scheme (Umlauf and Burchard, 2003). For lateral diffusion, only
        minimal eddy viscosity is applied because the horizontal resolution in AMM15 resolves the internal Rossby
        radius on the shelf. A bi-Laplacian diffusion scheme is used along model levels for both momentum and
tracers, with coefficients of $6 \times 10^{7}$ m$^{4}$ s$^{-1}$ and $1 \times 10^{5}$ m$^{4}$ s$^{-1}$, respectively. For more information on the core
        model configurations see Graham et al. (2018a). The data used in this study is extracted from the same hindcast
        simulation presented in (Graham et al., 2018b).





**2.5. Modelled particle tracking study**

To test the implication of the observed drifter trajectories, we conducted a series of offline particle tracking
experiments using AMM15 modelled velocities, with daily mean full-depth U/V velocities obtained from a
hindcast for the period of interest. From this dataset, horizontal velocities at 20 m and 70 m depth were
extracted to coincide with the TPM and drifter observation depths. We contrasted two periods sampled by the
drifters; 1st to 11th August 2013 and 15th to 25th December 2013. In each case, the 10-day interval was deemed
an 'observation period'. Particles were released at daily intervals across the local model domain for the 40 days
preceding the observation period (Fig. 3). Particles which were advected into the observation polygons during
the observation period were identified, and their release location noted. The experiment was repeated 5 times
resulting in a total of 200 unique particles being released from each location in Fig. 3a. Our analyses focus on
the origin of the particles which reached the observation polygons on the inner shelf.

The particles were tracked using a 2D Lagrangian scheme. The 2D location of a particle $X_p^t(x, y)$ at time $t$ is
calculated using

$$X_p^t(x, y) = X_p^{t-\Delta t}(x, y) + \Delta t. U_p(x, y) + \partial_H ,  \tag{1}$$

where $\Delta t$ is the model time step, $U_p(x, y)$ is the 2D model velocity at the particle location and $\partial_H$ is a 'random
walk' diffusive component (following Gillibrand and Willis (2007) and van Sebille et al., (2018)):

$$\partial_H(x, y) = \gamma [6. K_H. \Delta T]^{1/2} ,  \tag{2}$$

where $\gamma$ is a real random number ($\gamma \epsilon [-1, 1]$) and $K_H$ is the horizontal eddy diffusivity. For the 1.5 km model
grid of AMM15, a diffusivity of 1 m² s⁻¹ was chosen to reflect the low lateral momentum diffusion used in the
model physics.

**3. Results**

**3.1. Drifter study**

In this study only the on-shelf portion of drifter tracks (depth < 200 m) are investigated; for a detailed analysis
of shelf-edge dynamics from the drifter observations see Porter et al., (2018). The drifters were initially
released into the slope current but diverged at the canyon system at 55.5 °N with some moving offshore and
being advected southwards for a time before crossing the shelf edge, and others moving on-shelf north of the
release point. The periods of drifter transit of the Malin shelf can be split into 2 groups: the first group crossed
the shelf edge in August and September 2013, whereas the second group initially recirculated in deep water
before moving on-shelf in mid-December 2013. We use only tracks from drifters drogued at 70 m as this
enables a comparison between the behaviours of the autumn and winter groups. Shallow bathymetry prevents
the 70 m drifters from passing through Tiree Passage, but as flow on the outer Malin shelf is close to barotropic
in December (Davies and Xing, 2003; Ellett and Edwards, 1983; Jones et al., 2018) we consider these results to
be somewhat representative of near-surface currents.

Cross-shelf drifter progress in the complex Malin shelf region is dependent on local bathymetry and the location
each drifter crossed the shelf edge. However drifters from both groups were advected eastwards in the AIC and





passed through the same region north of Ireland, so the time taken to reach this point from the shelf edge
provides a metric of cross-shelf progress (Fig. 4). Beyond this point the tracks turn northward and scatter, and
inter-comparison of drifters once again is problematic. Consequently, we chose a meridional line at 8 °W
(roughly mid-shelf) to define drifters as having reached the inner shelf and by implication, being on a track
which passes near to or terminates at the Scottish coastline.

Seven drifters travelled between the shelf edge and 8 °W in the AIC during August / September 2018, and 2
drifters transited this route in December 2013. A visual comparison of tracks in Fig. 4 shows that drifter speeds
were typically between 10 and 20 cm s$^{-1}$ in group 1 (August 2013). One of the drifters in Fig. 4a briefly crossed
the shaded region delineating 70 m bathymetry without any indication of seabed interference in its strain gauge;
we speculate that there may be errors in the bathymetry in this region.

Flow at 70 m was generally on-shelf but with numerous sub-tidal meanders and reversals. Drifters crossing the
shelf edge between 55.4 °N and 56.1 °N were recruited into the AIC. This behaviour is contrasted by the drifters
which arrived on-shelf December 2013. Both drifters crossed the shelf edge at 54.75 °N and travelled in a
north-easterly direction at speeds of 20-60 cm s$^{-1}$. At 55.6 °N they turned eastward and took very similar paths
through the meridional line at 8 °W. While drifter speeds in August inferred a cross-shelf transport of 0.2 Sv
(Porter et al., 2018) the peak drifter speeds of 60 cm s$^{-1}$ observed in this study suggest that oceanic water import
via the AIC may briefly reach 0.48 Sv. This may be near the upper limit for transport of oceanic water towards
the coastline in the AIC, based on the long-term salinity and current observations at the TPM.

To appraise the meteorological and oceanographic conditions on the shelf during the experiment, we compared
the periods of drifter shelf transit with time series of reanalysis wind data on the Malin shelf, the 20 m current
speed at the TPM and the 20 m salinity at the TPM (Fig. 5). Note that the TPM was undergoing maintenance
during the early drifter experiment so TPM current speed and salinity data commence in late August 2013. The
cross-shelf transit time of drifters in group 1 (August 2013) is between 18 and 42 days. Wind forcing during
this period is typical for the time of year at this location, with numerous short episodes of SW-NW winds of 10-
15 m s$^{-1}$. In December 2013 however, three unusually strong westerly wind events occurred during a period
dominated by westerly airflow. We can see from Fig. 2 that events of this magnitude typically occur once or
twice per year during most winters. The onset of the first event on 5$^{th}$ December 2013 (Event A, Fig. 5)
precedes an increase in along channel currents at the TPM from ~5 cm s$^{-1}$ to greater than 30 cm s$^{-1}$ to poleward
(Event B). Salinity at the TPM increases abruptly from 34.5 to 35 on 14$^{th}$ December 2013 (Event C) and
remains almost continually above 34.75 until 30$^{th}$ December 2013. The drifters transiting the shelf during this
period do so in 6 and 10 days respectively.

### 3.2. Modelled particle tracking

The combination of drifter observations and the TPM salinity observations provides contrasting snapshots of
cross-shelf flow during intense storms and a more quiescent period. However, the observations are in some
ways not comparable: for example, when analysing the storm event, we compare the salinity at the TPM at 20
m depth, and the behaviour of drifters drogued at 70 m depth which could not pass through Tiree Passage due to
its shallow bathymetry. To present a more compelling picture of the periods under investigation, we supplement
these observations with a series of particle tracking experiments. Specifically, we seek to answer the question:



Did the episode of stormy weather in December 2013 significantly alter the origins of water reaching the Scottish west coast throughout the water column? To address this question, we performed a series of backward particle tracking experiments, focussing on particles terminating at the inner Malin shelf. The particles were released at 2 depths: 20 m (the nominal depth of TPM observations) and 70 m (the drogue depth of the deep
drifters) at the locations shown in Fig. 3a.

Figure 6 compares the tracks of particles terminating at the inner Malin shelf during August 2013 (the first episode of on-shelf drifter advection) and during December 2013 when there was rapid on-shelf advection of 2 further drifters. In August, most particles at both 20 m and 70 m originated within 50 km of the observation polygon, with a few cells west of 8 °W featuring greater than 20 % particle contribution. There is a small
contribution from the North Channel of the Irish Sea at 20 m, but this channel is closed to the 70 m particles by shallow bathymetry north of Ireland. The distribution of particles during the storm event (Fig. 6c and d) provides a contrasting picture, with up to 50 % of particles released in some regions west of 8 °W terminating in the observation polygon. At 20 m depth, there is a clear preference for oceanic water reaching Tiree Passage which is not present in the August experiment. The 70 m particles mostly originated from near the shelf edge at
55.5 °N during the storm event. While a minority of 20 m particles originated in the abyssal ocean and crossed the shelf edge during the storm event, at 70 m all tracked particles remained on the Northwest European shelf for the duration of the experiment.

A measure of typical particle advection times can be obtained by colouring particle origin cells by the average time their particles took to arrive at the observation polygons (Fig. 7). Note that Fig. 6 and Fig. 7 were produced
using separate (but identical) particle releases, and thus exhibit small differences in particle distribution due to the diffusive component of the particle motion. Again, the contrast between the 20 m particles in August, and those in December, highlights differences in the origins of the water. The source region of the December particles includes a broad section of the shelf edge whereas in August the waters are sourced exclusively from the shelf. In addition, the December distributions indicate a relatively rapid pathway between the shelf-edge at
55 °N and the observation polygons, such that particles originating at this location can be expected to arrive at the coastline in 20-25 days.

**4. Discussion**

In this study we examined a period of intense shelf sea flows driven by a cluster of low-pressure systems. The salinity and current speeds measured at the TPM during this event were amongst the highest measured by the
mooring during its multi-year occupation. The spike in salinity at TPM, coupled with the trajectories of two drogued drifters, confirmed that the origin of the water passing through Tiree Passage switched to the outer shelf during this period. In Section 4.1 we discuss the rapid (1-2 day) increase in currents in terms of the dynamic response to wind-induced pressure gradients on the Northwest European Shelf. This precedes the increase of salinity at the TPM by 8 days which we interpret as the time taken to advect high salinity water from a remote
location in the enhanced shelf currents. The nature of the high salinity intrusion is discussed in Section 4.2. We then consider the additional insight provided by the particle tracking in Section 4.3, and the weather conditions associated with HSPs in Section 4.4. In Section 4.5 we investigate the likelihood of an HSP occurring in each winter.





### 4.1. Dynamic response of the Malin shelf to wind forcing

There is strong evidence in the literature that wind-induced pressure gradients can quickly setup or enhance currents on the Malin shelf, and that their pathways are influenced by the wind direction (Davies and Xing, 2003; Inall et al., 2009; Jones, 2016; Xing and Davies, 2001). Similarly, the flow through the North Channel of the Irish Sea onto the Malin Shelf is highly correlated to wind aligned with the channel with a lag time of a few hours (Bowden and Hughes, 1961; Brown and Gmitrowicz, 1995). We may expect a rapid setup of inner shelf

currents in response to a wind-induced surface pressure gradient as the barotropic adjustment time will be limited only by the speed of a long wave in shallow water. This speed is set by $\sqrt{gh} \approx 30$ m s$^{-1}$ which equates to a shelf-wide effect within a few hours. In addition, an abrupt onset of flow will be subject to inertial effects, so that it may take one to two days for a stable flow to become established.

If we consider the AIC and Tiree Passage to be subject to the same forcing influences, one might expect a link

to exist between the speed of the drifters tracking the AIC and the poleward current speed concurrently measured at the TPM. However, we find little coherence between these measures (not shown), though both the drifters and the TPM do show increased current speed in December compared with August. We surmise that the instantaneous sub-tidal speed of the drifters is a complex aggregate of wind-driven currents and local bathymetric flow intensification, so their speed does not correlate simply with that measured at the fixed

mooring.

### 4.2. Advection of salt tracer

As a tracer of oceanic-origin water, the transport of salt is limited by the current speed on the shelf, and the 8-day lag between the increase in current speed and the onset of high salinity at the TPM points to the advection of the high salinity water from a remote source. This lag period is in accord with the time taken by the December

drifters to reach the inner shelf from the shelf edge (between 6 and 10 days).

Integrating the TPM poleward current flow over the interval between the onset of enhanced flow (6[th] December 2013, Event B) and the arrival of high salinity water at the TPM (Event C, 14[th] December 2013) gives a total displacement of 145 km. If we assume this water arrived in Tiree Passage via the AIC, the minimum distance from the shelf edge to the TPM via this route is 210 km, and the December drifters in fact crossed onto the shelf

further south (54.75 °N, Fig. 4b) necessitating travel of at least 270 km to reach the TPM from the shelf edge at this location. It therefore seems likely that high salinity water was already residing on the shelf prior to the first storm on 5[th] December 2013. The presence of near-oceanic water on the outer shelf is supported by observations of salinity distribution (Jones et al., 2018; Jones, 2016), radioisotopes tracking coastal water extent (McKay et al., 1986; McKinley et al., 1981) and distribution of other water properties such as temperature,

chlorophyll and nutrients on the shelf (Ellett, 1979; Ellett and Edwards, 1983; Siemering et al., 2016). In order to produce the observed lag at the TPM, we estimate that the high salinity water originated at approximately 7.9 °W at the latitude of the AIC, which is the location of maximum particle origin percentages in Fig. 6c.

Unlike salinity, the temperature measured at the TPM (not shown) does not exhibit any notable deviation from the expected seasonal cooling at the time of the HSP investigated here. This is because there is very little

difference between oceanic (near surface) and coastal water temperatures on the Malin shelf in December (Ellett



and Edwards, 1983; Inall et al., 2009; Jones et al., 2018). A change in water origins from coastal to oceanic, however abrupt, would therefore have little impact on water temperatures at this time. However, between January and March coastal waters are cooler (6-8 °C) than the adjacent ocean (9-10 °C) so we would expect a similar event during this period to increase coastal water temperatures in western Scotland.

### 4.3. Particle release

The model particle releases demonstrate a striking contrast between shelf behaviour in August 2013 and that during the storm event in December 2013. As might be expected from the evidence thus far, current speeds were slower in August and particles in the observation polygons originated more locally over the 50-day tracking period. This was true both for particles in the seasonal thermocline (20 m) and below it (70 m). By contrast, most particles tracked during the storm event originated between the mid-shelf and the shelf edge. The long transit time of the August particles would be associated with greater mixing between oceanic and shelf waters before reaching the coast. We would therefore expect that the high salinity associated with the inflow would be more dilute than during December. The preferential across-shelf pathway taken by most particles in Figure 6d closely matches the route taken by the two drifters in Fig. 4b. This supports the notion of the AIC as a narrow, jet-like current which was proposed by Porter et al., (2018). Further clarity on the modelled flow is obtained by comparing the average modelled current speed during the two 10-day observation periods (Fig. 8). The key difference between the August and December observation periods is the rapid across-shelf flow in December, which is not present during August. This disparity illustrates why most particle recruitment in August was restricted to the inner shelf: there was no pathway to enable transport from the shelf edge. In December, on-shelf flow exceeded 0.3 m s$^{-1}$ in much of the AIC at 20 m depth and the pattern is similar, though slightly weaker, at 70 m depth.

Due to the design of the release and observation periods we might expect about 1 in 5 local releases to result in an observation in the polygons given steady flow. This is because particles were released over a period of 50 days, but were only tallied in the observation polygons for the final 10 days of the experiment (Fig. 3b) so the majority of local particles would pass through the observation polygon without being counted. However, in the December experiments we see high concentrations of particles originating remotely. This is a result of flow speeds increasing during the simulation, resulting in an effective convergence of particles during the observation period (days 40-50).

At the shelf edge, the insulating effect of the steep bathymetry maintains a strong control on ocean-shelf interaction for the model particles. There are almost no examples of 70 m particles crossing the shelf edge and only a small proportion of 20 m particles crossed from ocean to shelf. Graham et al. (2018b) reported an on-shelf volume flux of roughly 5 x 10$^4$ m$^3$ s$^{-1}$ per 60 km shelf-edge in this region which is higher than most other sections of the European shelf edge. Given the evidence presented here for mid-shelf flows perhaps an order of magnitude greater during the stormy period, it is unsurprising that particles appear to be recruited from a broad swathe of the shelf edge before converging in the AIC. We find some preference for the recruitment of particles at the canyon systems around 55 °N and 55.5 °N (Fig. 7) in agreement with the behaviour of the drogued drifters (Porter et al., 2018). These canyons appear to cause a breakdown in slope current stability leading to greater

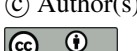



cross-slope flow in their vicinity. For further examination of the implications of drifter trajectories on ocean-shelf exchange in this region, see Burrows and Thorpe (1999) and Porter et al., (2018).

Graham et al. (2018b) found that cross-shelf fluxes in the surface layer (0-20 m) were larger during winter (January) than in the summer (July). The enhanced cross-slope flow during winter is likely to be due to increased wind forcing driving a downwelling circulation on the shelf, as described by Holt et al. (2009). The storm event depicted in the present study can be regarded as a snapshot of the processes contributing to the long-term winter average.

**4.4. Prerequisite weather conditions for a high salinity pulse on the west coast of Scotland**

Given the regular passage of low-pressure systems across the UK during most winters, it is perhaps surprising that only 1-2 HSPs per year are recorded by the TPM. This finding indicates that further prerequisites are required to transport oceanic water to this inner shelf location. There is much evidence that the Malin shelf is subject to an underlying baroclinic flow originating in the Irish Sea; from tracers, oceanographic observations

and rotating tank experiments (Ellett, 1979; Ellett and Edwards, 1983; Hill, 1987; Jones et al., 2018; McKay et al., 1986; McKay and Baxter, 1985). Jones et al., (2018) conjectured that the relative infrequency of HSPs was because the inner shelf had a tendency to return to this baroclinic state in the absence of wind forcing. They argued that the displacement of water driven by a single low-pressure system may not be sufficient to advect oceanic water across the 'buffer' of coastal waters which typically occupy the inner shelf. If we use for example

the approximate linear relationship between wind and current speed in shallow seas suggested by Whitney and Garvine, (2005):

$$U_{wind} \approx 2.65 \times 10^{-2} U_{10} , \qquad\qquad\qquad (3)$$

where $U_{10}$ is the 10 m wind speed and $U_{wind}$ is the approximate developed wind induced current flow, a fully-developed current of 47 cm s⁻¹ would be expected to result from a 24 hour mean wind speed of 18 m s⁻¹. This

equates to a 24-hour displacement of ~40 km which is substantially less than the estimated 145 km travelled by the body of high salinity water observed at the TPM. The additional factor prior to the December 2013 HSP may be that the storm on the 5th December was followed by several days of strong southerly and westerly winds (e.g. Fig. 5b) which sustained the current system long enough for the outer shelf water to arrive in Tiree Passage 8 days later. The strong flow towards the coast and thus high coastal salinities were maintained throughout most

of December 2013 by a pattern of low-pressure systems passing over Scotland. Jones et al (2018) found surprisingly low correlations between southerly and westerly winds and the salinity measured at TPM, and we suggest their finding was caused by this non-linear 'pumping' of oceanic water to the inner shelf. The trade-off between cumulative storm frequency, length scales and baroclinic flow strength is likely to dictate the varying response to wind reported for other coastal current systems (e.g. Brooks and Townsend, 1989; Wiseman et al.,

1997; Lentz et al., 2006)

Strong on-shelf flow towards the Scottish west coast only occurs during southerly to westerly winds, as other wind directions either drive Irish Sea water onto the shelf or retard shelf flow more generally (Davies and Xing, 2003; Jones et al., 2018). To illustrate the relevance of this note to the present study, sea level pressure maps of the major wind events associated with the December 2013 HSP are shown in Fig. 9.



In each case an area of intense low pressure passed to the north of the UK, resulting in a westerly airflow over
the Malin shelf. The path of low-pressure systems over the North-eastern Atlantic is captured by the North
Atlantic Oscillation (NAO) index (Hurrell, 1995) which compares the atmospheric pressure over Iceland and the
Azores. Positive NAO winters such as that of 2013/14 feature a northerly storm track, whereas during negative
NAO years low pressure systems tend to pass south of the UK. Consequently, conditions likely to result in an
HSP episode are more likely to occur during positive NAO years. In the following section we explore the
relationship between the NAO and storminess over the North-eastern Atlantic.

### 4.5. North Atlantic Gale Index and its relationship with the NAO

We now analyse wintertime storminess over the North Atlantic by using daily 10m windspeed data from the
ERA-Interim 0.75° x 0.75° reanalysis product. Given this spatial and temporal resolution, peak wind speeds are
likely to be underestimated, but the low-pressure systems of interest to this study (as illustrated in Fig. 9) are
well resolved by the ERA-interim product. Furthermore, the AMM15 model used in the Lagrangian trajectory
analyses was forced using ERA-Interim data.

Following Qian and Saunders (2003) we define a set of wind speed indices as the numbers of days in each
Winter (DJFM) that wind speeds exceed different force levels on the Beaufort Wind Scale/World
Meteorological Organisation (WMO) windspeed classification. These levels are: near-gale force (Beaufort Scale
7, 13.9–17.1 ms$^{-1}$), gale force (Beaufort Scale 8, 17.2–20.7 ms$^{-1}$), strong gale force (Beaufort Scale 9, 20.8–24.4
ms$^{-1}$), and storm force (Beaufort Scale 10, 24.5–28.4 ms$^{-1}$). One 'day' is counted for a particular grid cell and
Beaufort Scale level if the windspeed within that grid cell exceeds the lower limit of the particular scale level for
that day. The total number of 'days' is then accumulated for each grid cell, Beaufort Scale level, and winter
(DJFM) season. The winter is defined as the year containing JFM, i.e. for a winter December 2013 – March
2014, the year is 2014.

Analysing the period 1979 to 2015 we perform an EOF decomposition of "gale force" winds over the whole
North Atlantic (i.e. 17.2 ms$^{-1}$ Beaufort scale 8, or greater). The first EOF mode (Fig. 10) captures 49% of the
total variance. The mode 1 time series, which we define now as the Gale Index, correlates significantly with the
winter NAO index over the same period, with r = 0.84 (Fig. 11).

The mode 1 EOF pattern (Fig. 10) shows the storm track with a WSW to ENE orientation, as is typically
associated with NAO-High winters. The most energetic landfall of this pattern is between 56°N and 58°N,
broadly our region of interest. We note that during the winters of 2004, 2010 and 2013 no HSPs were recorded
in TPM record (Fig. 2), coinciding with below-average Gale Index. In 2006 the Gale Index was also low, but no
winter TPM data exist for that winter. 2004 and 2006 are interesting years because the Gale Index and NAO
Index exhibit differing behaviour in those years, suggesting the Gale Index may be a better predictor of shelf
salinity conditions or HSPs than the NAO-Index, despite the very high correlation between the two indices (r =
0.84).

Between 1989 and 1994 Fig. 14 of Inall et al. (2009) shows a prolonged period of more saline shelf waters at
10m depth from Ellett Line CTD stations. This period corresponds to a high Gale Index period in the EOF mode
1 eigen value time series (Fig. 11). Further, the relative dip in the Gale Index in winter 1991 apparently
coincides with a modest westward relaxation of isohaline contours, with fresher waters reappearing east of





6.5°W. This 1989 to 1994 period precedes salinity measurements on TPM, so we cannot comment on HSPs during this period of particularly stormy winters.

**5. Conclusions**

In this paper we have characterised an oceanic inflow onto the Northwest European shelf during intense storm activity using drifters, a moored time series and modelled particle tracking.

We found that during the storm event, across-shelf flow increased from 10-20 cm s$^{-1}$ to 60 cm s$^{-1}$, with a commensurate increase in estimated shoreward transport from 0.2 Sv to 0.48 Sv. Furthermore, we linked this unusually high import of oceanic water with an abrupt increase in salinity at the coastal mooring (from 34.5 to 34.85 in 24 hours) which occurred 8 days after the onset of storm activity. Given the lag time and the path of the drifters, we deduced that the coastal salinity spike was caused by the rapid import of water situated on the mid-shelf, at roughly 55.6 °N, 7.9° W prior to the current intensification. Using modelled particle tracking, we showed that coastal water was preferentially recruited from the mid-to-outer shelf during the storm event, both at 20 m (the depth of the moored time series) and at 70 m (the depth of the drogued drifters). This contrasted with a more typical quiescent period in which coastal water was recruited locally over the experiment duration.

The spike in coastal salinity associated with storm activity was one of several in the 13-year mooring time series. To assess the likelihood of these sporadic storm episodes occurring during a given winter, we constructed a 'Gale Index'; a measure of the number of days wind speeds exceeded predefined thresholds across the North Atlantic. Despite high correlation between the Gale Index and the NAO-Index (r = 0.84), we found the Gale Index may be a better predictor of salinity observed on the shelf than the NAO-Index.

**6. Data availability**

AMM15 model data are archived on the Met Office mass storage system and can be accessed through the STFC-CEDA platform JASMIN (Lawrence et al., 2013). Access to NEMO code and parameterizations used are outlined in Graham et al., (2018a). The drogued drifter data and Tiree Passage Mooring data are available from the British Oceanographic Data Centre (www.bodc.ac.uk).

**7. Author contribution**

SJ prepared the manuscript with contributions from all co-authors. MI secured the funding for the work, developed the WMO gale index (Sec. 4.5) and made significant contributions to the particle tracking experiment design. MP Designed and executed the drogued drifter release experiment and help to shape the present study. JG was a lead contributor to the design and testing of the AMM15 model and was instrumental in ensuring the particle tracker was appropriately tuned for model diffusivity. FC helped to secure the funding for the work, provided mentorship in the form of PhD supervision to SJ and identified the relation between drifter behaviour and HSPs in the mooring salinity record.

**8. Competing interests**

The authors declare that they have no conflict of interest.



**9. Acknowledgements**

This study was funded by the UK Natural Environment Research Council (NERC) projects Fluxes Across
Sloping Topography of the North East Atlantic (FASTNEt) (NE/I030151/1) and Overturning in the Subpolar

North Atlantic Program (OSNAP) (NE/K010700/1). MP was supported by the AtlantOS project (European
Union's Horizon 2020 research and innovation program, grant: 633211). The authors would like to thank
colleagues at the Met Office for their assistance in providing access to archived model output.

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

**Figures**

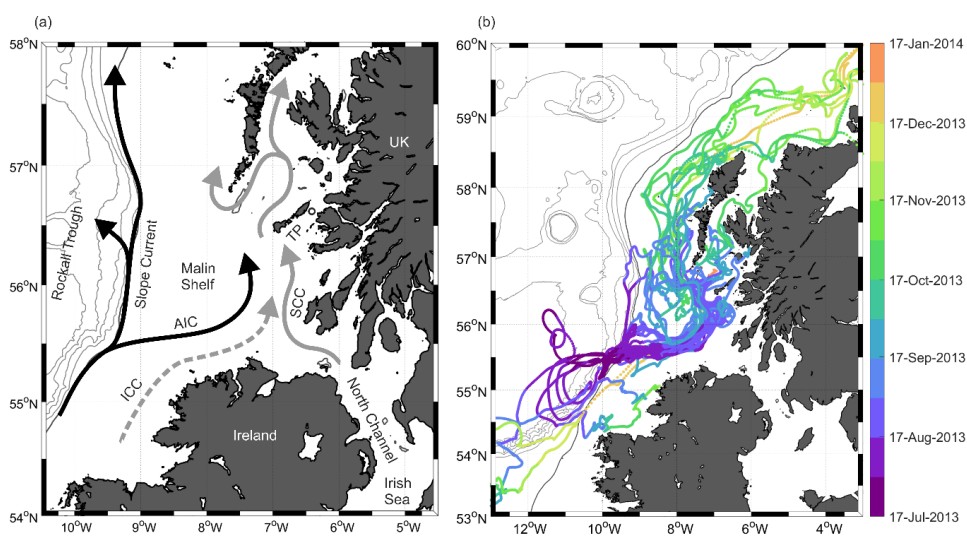

Figure 1: a) Schematic of study regions showing features mentioned in text. Oceanic water pathways shown in

black, coastal water pathways shown in grey. Solid grey line shows the Scottish Coastal Current (SCC), dashed

grey line indicates the pathway of the Irish Coastal Current (ICC, summer only). AIC: Atlantic Inflow Current,

TP: Tiree Passage. The location of Tiree Passage Mooring is indicated by the black circle. b) Trajectories of

drifters released during FASTNEt JC88 cruise (drogued at 15 m and 70 m), coloured by date. Bathymetry

contours from GEBCO bathymetry (http://www.gebco.net/). GEBCO = General Bathymetry Chart of the

Oceans.

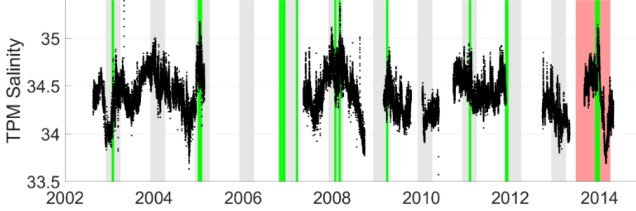

Figure 2: Salinity at the Tiree Passage Mooring, at 20 m. Grey bars denote winter months (DJFM). Green lines

show instances of daily mean westerly winds on Malin shelf exceeding 18 m s$^{-1}$ after Jones et al., (2018), there

are 17 occurrences between 2002 and 2015. Pink bar highlights the drifter study period (shown again in Fig. 5).



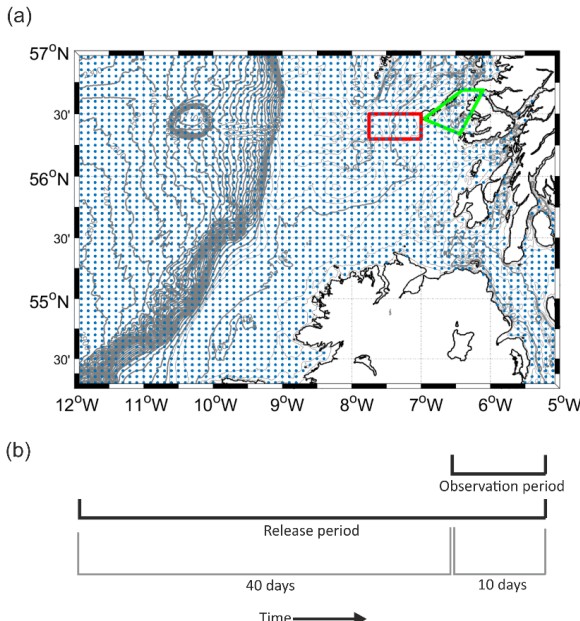

Figure 3: a) Map of particle release locations, green polygon shows observation region for 20 m particles; red polygon shows observation region for 70 m particles; b) schematic showing observation period preceded by 40 days of particle releases. Bathymetry contours from AMM15 model bathymetry.


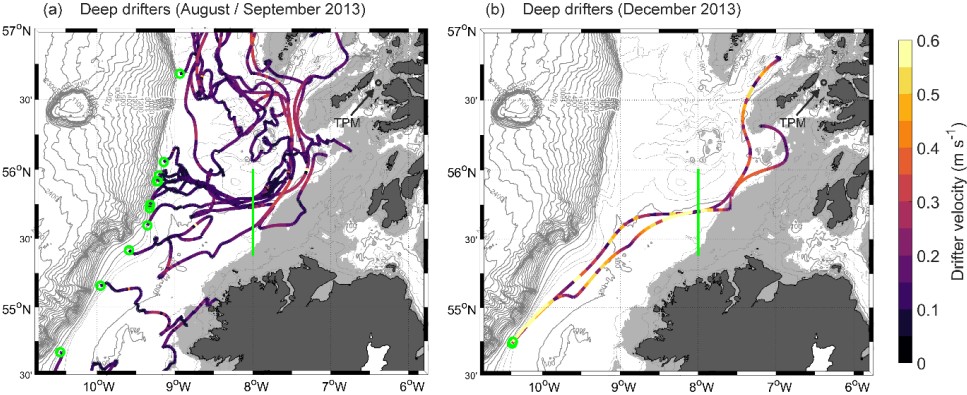

Figure 4: Trajectories of deep (drogued at 70 m) drifters, coloured by velocity and separated into drifters which travelled on-shelf in a) August-September 2013 and b) December 2013. The tracks are set to commence when each drifter first crossed the 200 m isobath (beginning of each track indicated by green circles). Green line at 8 610 °W indicates where drifters are deemed to have reached the inner shelf. Pale shaded region delineates 70 m bathymetry. Bathymetry contours from GEBCO bathymetry (http://www.gebco.net/).



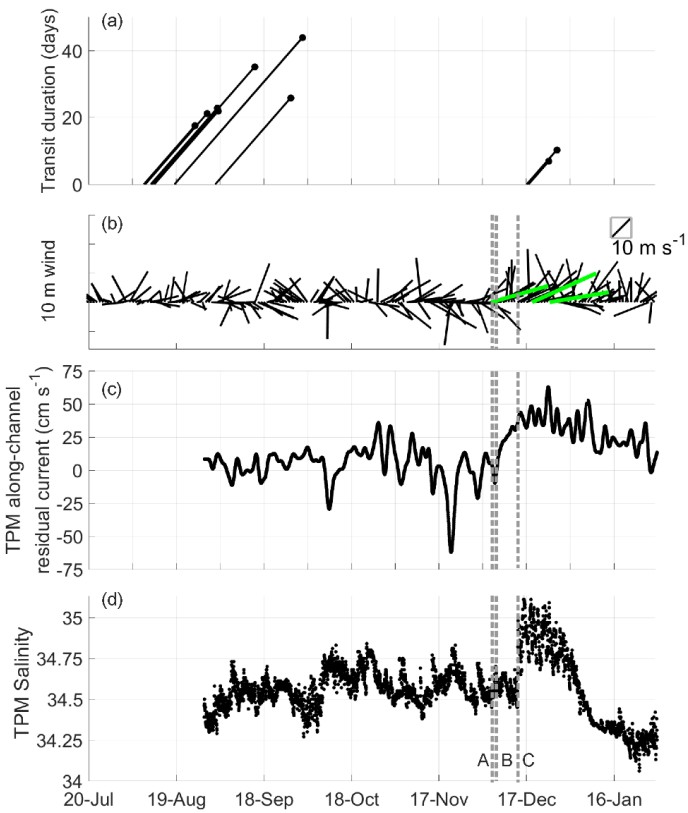

Figure 5: a) Transit times and durations of 70 m drifters between the shelf edge (200 m isobath) and a meridional line at 8 °W signifying arrival at the inner shelf. Lines commence as each drifter crosses the shelf edge and terminate when it passes 8 °W. b) Vector plot of 10 m wind speed and direction on the Malin shelf (56 °N, 7 °W) derived from ECMWF ERA-Interim product. Instances of westerly storm events (daily mean westerly wind > 18 m s⁻¹) are shown in green. c) Low-pass filtered along-channel currents at TPM, d) 20 m salinity at TPM. Grey dashed lines denote the following events: A: first storm event. B: Onset of strong poleward currents at TPM. C: Beginning of high salinity pulse (HSP) at TPM.






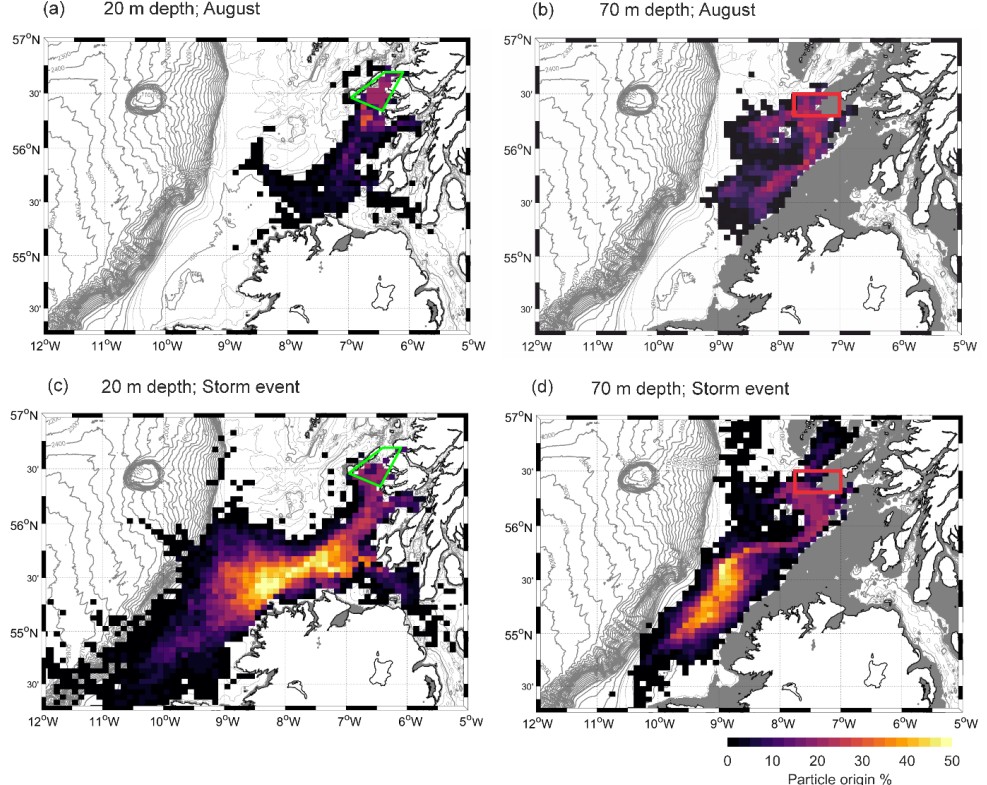

Figure 6: Particle distribution maps showing the origin of particles advected through Tiree Passage and the inner

Malin shelf. Green polygon shows the observation region for the 20 m particles in a) and c); for the 70 m

particles in b) and d) the polygon (red) instead bisects the tracks of the drogued drifters as Tiree Passage is not

open at this depth. Each cell is coloured by the percentage of particles released at that location which were

subsequently advected through the observation region. The observation period for a) and b) is 1[st] to 11[th] August

2013, for c) and d) is 15[th] to 25[th] December 2013. Note that these maps show particle release locations only and

are not representative of model resolution. Bathymetry contours from GEBCO bathymetry

(http://www.gebco.net/).

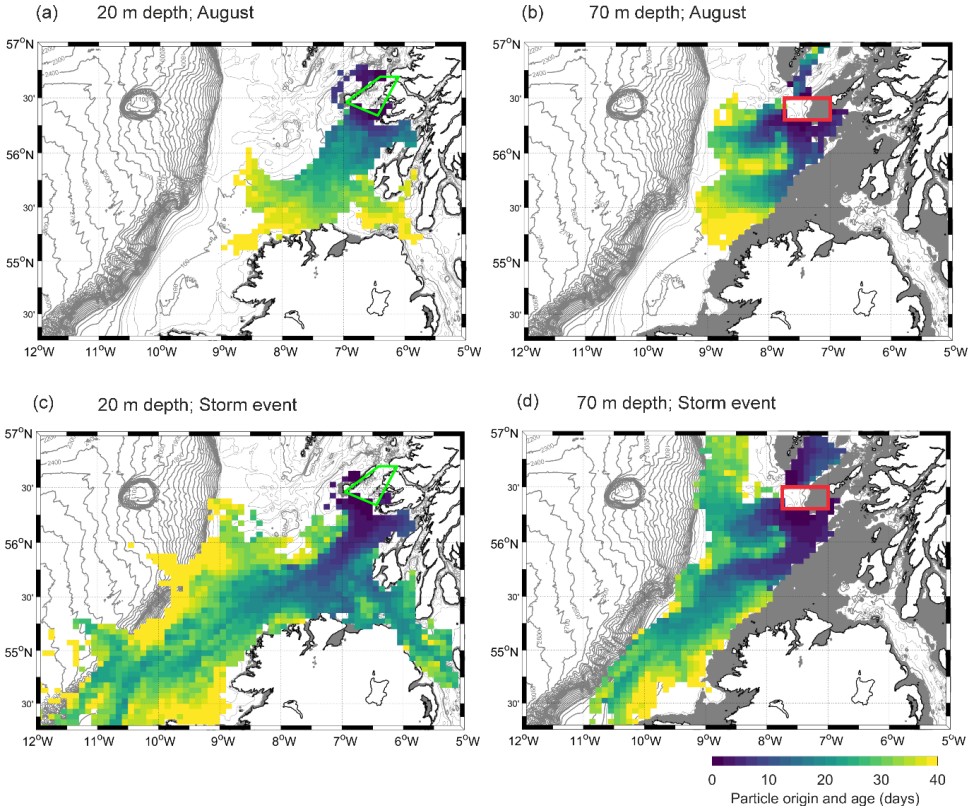

Figure 7: Particle distribution maps showing the origin of particles advected through Tiree Passage and the inner Malin shelf.  Green polygon shows the observation region for the 20 m particles in a) and c); for the 70 m particles in b) and d) the polygon (red) instead bisects the tracks of the drogued drifters as Tiree Passage is not open at this depth.  Each cell is coloured by the average age of particles originating at this location, i.e. the number of days between the release of a particle and its arrival in the observation region.  The observation period for a) and b) is 1$^{st}$ to 11$^{th}$ August 2013, for c) and d) is 15$^{th}$ to 25$^{th}$ December 2013.  Note that these maps show particle release locations only and are not representative of model resolution.  Bathymetry contours from GEBCO bathymetry (http://www.gebco.net/).





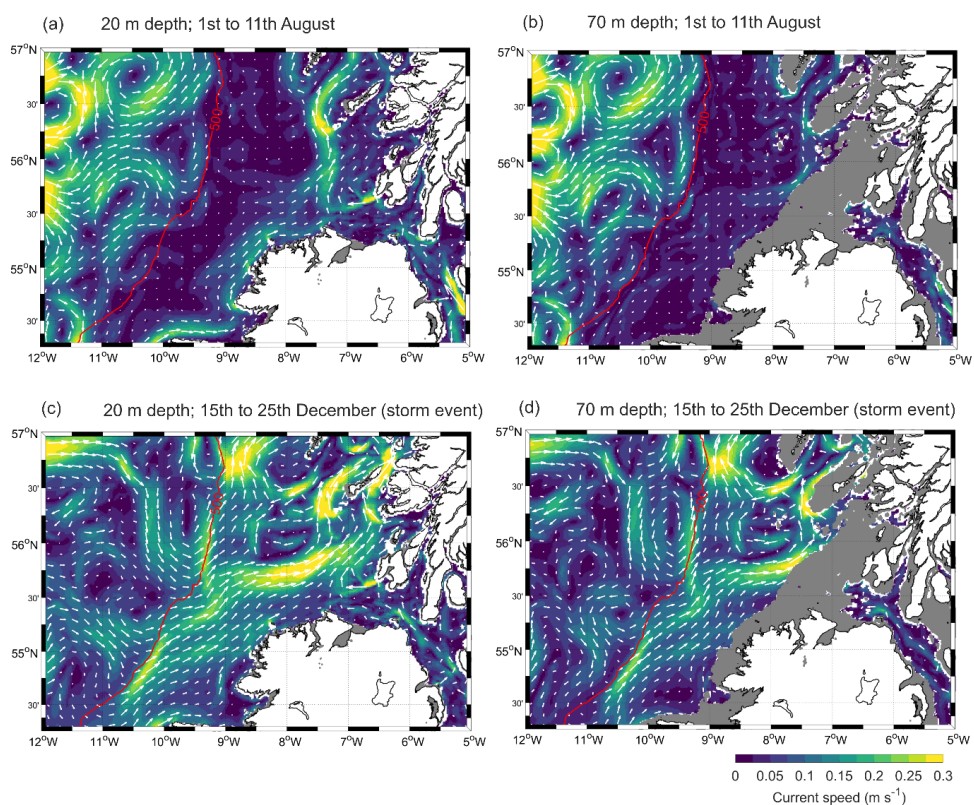


Figure 8: Average horizontal current velocity in AMM15 model for: a) 1st -11th August 2013 at 20 m depth, b) 1st-11th August 2013 at 70 m depth, c) 15th-25th December 2013 at 20 m depth, d) 15th-25th December 2013 at 70 m depth. Bathymetry contours from GEBCO bathymetry (http://www.gebco.net/).





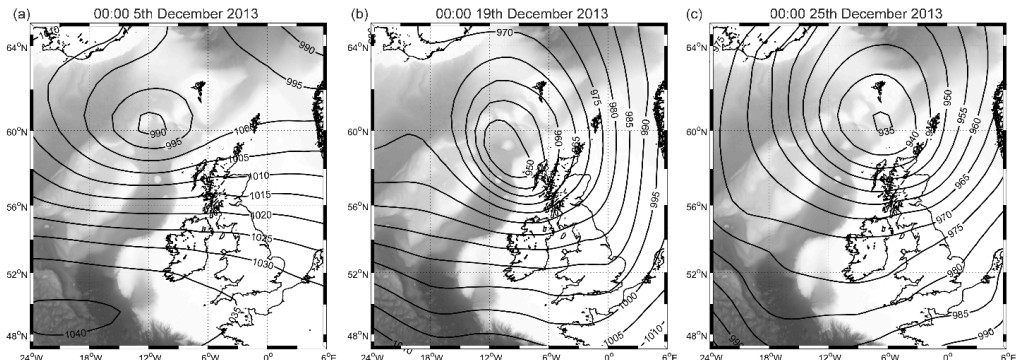

Figure 9: Sea Level Pressure maps for the 3 storm events associated with the winter 2013-14 high salinity pulse (HSP). Sea level pressure data derived from ECMWF ERA-Interim 0.75° x 0.75° product. Bathymetry contours from GEBCO bathymetry (http://www.gebco.net/).


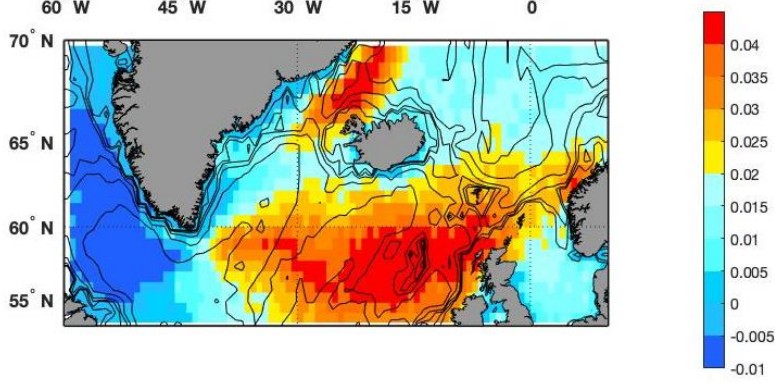

Figure 10: Mode 1 EOF Winter (DJFM) gale-days for North Atlantic. Based on ERA-Interim daily averaged 10m wind speed, and WMO gale definition of wind speed exceeding 17.2 ms⁻¹. Bathymetry contours from GEBCO bathymetry (http://www.gebco.net/).






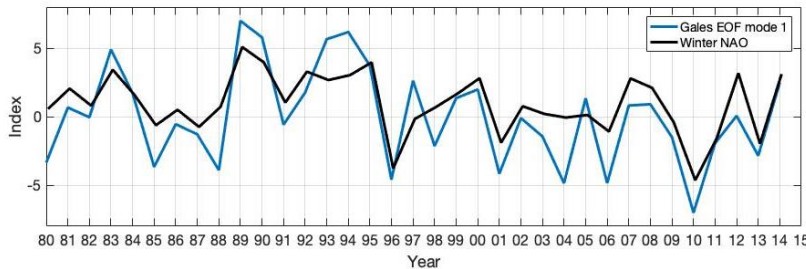

Figure 11: Time series (eigen values) of EOF Mode 1 eigen vector pattern illustrated in Figure 10 (blue line), defined as the North Atlantic Gale Index. NAO-Index after Hurrell (1995). The winter is defined as the year containing JFM, i.e. for a winter December 2013 – March 2014, the year is 2014. The two indices correlate

with R=0.84