# Peer review of "Storm-driven across-shelf oceanic flows into coastal waters"

_Ocean Science, 2019_

## Referee Comment (RC1) · Anonymous Referee #1 · 4 Dec 2019

General Comments

The authors present a range of observational and modelling evidence for a high salinity pulse (HSP) on the Malin shelf in December 2013, and further supporting evidence for HSPs in other winters, driven by persistent westerly gales generally associated with positive NAO states. These HSPs likely amount to a major oceanic flux into this sector of the northwest European shelf, with potentially wide-ranging consequences for nutrient levels and productivity in coastal waters.

In summary, the manuscript is well written and figures are of a high quality. It should be suitable for publication in Ocean Science, subject to minor revisions and/or appropriate responses, in regard to the specific comments listed below.

[Figure]

Specific Comments

1. p.4, lines 121-122: Regarding "Gaps in positional data were linearly interpolated with a maximum gap size of 20 hours" - can you provide information on the extent of this?

2. p.4, line 151: regarding the hindcast, in addition to citing Graham et al. (2018b), can you specify the period of hindcast, obviously covering August-December 2013.

3. 161-162: Regarding "The experiment was repeated 5 times resulting in a total of 200 unique particles being released from each location in Fig. 3a.", how was each experiment different? Was this simply sampling a range of 'random walks'?

4. p.5, line 167: A few more details on the Lagrangian method would be appropriate. How do you interpolate in latitude and longitude to obtain model velocity at an arbitrary particle location? You refer to the model timestep Delta_t in Equation 1. How long is this? To which model do you refer? Presumably the Lagrangian scheme? Noting previous reference to offline particle tracking and use of daily mean modelled velocities, does Delta_t = 1 day, or much shorter? If the latter, how do you interpolate in time?

5. p.5, line 170: What is Delta_T? Presumably Delta_t?

6. p.5: Given the complexity of your study region, does the Lagrangian scheme involve any issues near bathymetry or coasts? Do particles "crash" into coasts or seabed if the timestep is too long?

7. p.5, lines 181-182: "We use only tracks from drifters drogued at 70 m as this enables a comparison between the behaviours of the autumn and winter groups." It is not clear to me what you mean - can you elaborate?

8. p.6, lines 203-204: Regarding ". . . the peak drifter speeds of 60 cm/s observed in this study suggest that oceanic water import via the AIC may briefly reach 0.48 Sv", how did you estimate this transport? Presumably by associating the speed with a cross-sectional area? Can you provide some detail? Perhaps briefly reiterate the method of

Porter et al. (2018)

9. p.7, line 231: As you introduce the particle statistics and Figure 6, can you briefly explain the particle origin (%) diagnostic in the text, as you have in the figure caption? Also, in addition to Figures 6 and 7, could you also show the mean particle salinity, as you have shown origin and age? Might this further confirm the pathway for high salinity water towards the Tiree Passage?

10. p.7, line 240: Regarding the statement "While a minority of 20 m particles originated in the abyssal ocean", do you mean that these particles upwelled from off the shelf? The abyssal ocean seems an exaggeration in this context

11. p.7, lines 257-258: Regarding "the rapid (1-2 day) increase in currents in terms of the dynamic response to wind-induced pressure gradients", I assume you mean geostrophic flows supported by a change in sea surface slope?

12. p.8, first paragraph: Continuing this theme, you suggest rapid setup on a short barotropic adjustment time, perhaps disrupted by inertial effects, but what about Ekman dynamics? Might one expect a strong westerly wind to drive Ekman drift towards Northern Ireland, setting up local downwelling and a geostrophic jet to the west, in the same sense as the AIC (or ICC in Fig. 1a)? Does this complement on-shelf oceanic flows?

13. p.9, lines 302-304: Regarding "between January and March coastal waters are cooler (6-8 °C) than the adjacent ocean (9-10 °C) so we would expect a similar event during this period to increase coastal water temperatures in western Scotland", is there evidence for this in the TPM temperature record?

---

## Referee Comment (RC2) · Anonymous Referee #2 · 8 Jan 2020

In "Storm-driven across-shelf oceanic flows into coastal waters", Jones and colleagues investigate the occurrence of High Salinity Pulses at a coastal location on the Scottish continental shelf. The work builds on drifter deployments from the FASTNEt research project, a sustained mooring/coastal observatory and numerical model simulations combined with particle tracking. Although the results rely on observations from two drifter tracks, they are well supported by information from the mooring and numerical model, suggesting the results are robust. The authors elaborate their analysis to investigate the occurrence and drivers of these HSPs in a longer time series record, linking to NAO and gale force wind variability.

This work is a valuable contribution on the subject of ocean-shelf exchange and the implications for conditions in shelf seas.

[Figure]

The manuscript is well written and structured, and I would recommend publication. I have some queries below, which I think are relatively straightforward to address and will, in my opinion, improve the overall manuscript.

L101-104: What is the weight we can attach to two drifter observations? The region has been extensively studied using drifter deployments. Are there other historic records that provide context to the occurrence of such a transport pathway? Further in the MS, the authors use a numerical model to provide context, but I wonder if the historic observational record should also be further explored.

L159-161 & Figure 3: Are these the blue points in Figure 3a? Please edit the caption of Figure 3a appropriately.

L161-162: Why was the experiment repeated five times? Is this to provide more tracks with a CPU-manageable method, or did the five different experiments have different parameters?

L169: Is T in Equation 2 the same as t in Equation 1?

L170-172: How sensitive are these results to the chosen horizontal eddy diffusivity?

L202-205: Where does off-shelf transport occur? Does this transport contribute fully to the pathway which leads into the North Sea?

L212-218 and Figure 9: Although not yet part of the UK Met Office/Met Eirean storm naming, these winter storms were named. I would recommend including their name as part of the text and Figure 9, as these are often used in other analyses: Xaver (5-6 December), Bernd (18-19 December) and Dirk (23-24 December).

L226-230: The choice of words here ("backward particle tracking experiments") is confusing. From my understanding, the authors performed a particle tracking experiment where particles were released from a source, and only those tracks which reached the observation location during the observation period were further analysed (see L162-163).

L244-246: Why were two different experiments used? Are these two from the five mentioned on L161-162? Or are these two times an experiment with 5 particle tracking simulations?

L302-304: Does the TPM record show such occurrences, i.e. where the temperature also changes in line with the salinity change? Figure 2 shows gaps in the salinity record at TPM, are more temperature data available which would then potentially allow an analysis of HSP-like events in the temperature record? On L413-414 there is a suggestion that the TPM had temperature data prior to 1994, which could be analysed in such a manner.

L346-347, Figure 2 and L253: If a cluster of low-pressure systems is a pre-requisite for the occurrence, are HSPs a winter-only phenomenon? How many HSPs occur outside the winter season? Could Figure 2 be edited so the identified HSPs are also plotted on (maybe a marker along the 33.5 or 35.5 salinity level)? I would also recommend changing L253 to say "winter storms".

Bathymetry contours in figures: The model bathymetry is based on the EmodNet data product. I would suggest contours based on this product in all plots.

Figure 2: In the discussions PDF, this figure didn't occupy the full width. This could be due to the editorial system, but I would recommend for the authors to check there is no unnecessary white space in the image. I think this figure merits a full A4 width space, to make sure it is legible.

Figure 8, caption: please add "(in red)" after "Bathymetry contours". As far as I could tell, this was also only the 500 m one, so I would suggest "Bathymetry contour (red) ... "

---

## Author Comment (AC1) · 5 Feb 2020

>1. p.4, lines 121-122: Regarding "Gaps in positional data were linearly interpolated with a maximum gap size of 20 hours" - can you provide information on the extent of this?

Added clarification in text (p4 line 123); drifters were set to report position every 3 hours. While instances of a single position being missed were relatively frequent, longer gaps were rare.

>2. p.4, line 151: regarding the hindcast, in addition to citing Graham et al. (2018b), can you specify the period of hindcast, obviously covering August-December 2013.

Change made (p4 line 154).

[Figure]

>3. 161-162: Regarding "The experiment was repeated 5 times resulting in a total of 200 unique particles being released from each location in Fig. 3a.", how was each experiment different? Was this simply sampling a range of 'random walks'?

That is correct, we re-ran the experiment several times to sample a range of diffusive random walks. Clarification added in text (p5 line 164).

>4. p.5, line 167: A few more details on the Lagrangian method would be appropriate.

Added more information as suggested (p5 lines 167 - 177). Details below.

> How do you interpolate in latitude and longitude to obtain model velocity at an arbitrary particle location?

The model velocity is obtained at the particle location by converting the model grid nodes into distance space and performing a bilinear interpolation (line 171).

> You refer to the model timestep Delta_t in Equation 1. How long is this? To which model do you refer? Noting previous reference to offline particle tracking and use of daily mean modelled velocities, does Delta_t = 1 day, or much shorter? If the latter, how do you interpolate in time?

We refer to the offline AMM15 output. Delta_t is thus 1 day or 86400 seconds in this context. We do not interpolate in time. See also response to Q. 6 regarding particle crashes.

>5. p.5, line 170: What is Delta_T? Presumably Delta_t?

Correct, modified in text for clarity (p5 line 170).

>6. p.5: Given the complexity of your study region, does the Lagrangian scheme involve any issues near bathymetry or coasts? Do particles "crash" into coasts or seabed if the timestep is too long?

Our primary interest was in the large-scale trends in the particle motion, particularly

the contrast between summer and winter-storm shelf behaviour. While the Lagrangian scheme with a 1-day timestep does introduce cumulative errors leading to occasional terrain crashes, we do not consider this to impact our conclusions. Fig. 1 shows comparisons between particle tracks in daily mean modelled velocities and hourly modelled velocities. While there are differences between particle trajectories in hourly and daily flow fields, the regions swept by the particles are very similar. We therefore feel that daily flow fields are adequate for this study.

>7. p.5, lines 181-182: "We use only tracks from drifters drogued at 70 m as this enables a comparison between the behaviours of the autumn and winter groups." It is not clear to me what you mean - can you elaborate?

15 drifters were also released with drogues at 15 m depth, but these all moved on-shelf in August 2013 so would not contribute to an autumn–winter comparison. Modified text to clarify (p5 line 181).

>8. p.6, lines 203-204: Regarding ": : : the peak drifter speeds of 60 cm/s observed in this study suggest that oceanic water import via the AIC may briefly reach 0.48 Sv", how did you estimate this transport? Presumably by associating the speed with a crosssectional area? Can you provide some detail? Perhaps briefly reiterate the method of Porter et al. (2018)

Porter et al. (2018) estimated this transport using the cross-sectional area of the current core derived from glider sections. The T and S thresholds of Eastern North Atlantic Water during this period ($10 - 10.5\,°C$, S >35.43) were chosen to represent the ingress of Atlantic water in the current. We assumed the same cross-sectional area but with revised average drifter speed to estimate the December transport. Added detail in text (p6 line 207-211).

>9. p.7, line 231: As you introduce the particle statistics and Figure 6, can you briefly explain the particle origin (%) diagnostic in the text, as you have in the figure caption? Also, in addition to Figures 6 and 7, could you also show the mean particle salinity, as

you have shown origin and age? Might this further confirm the pathway for high salinity water towards the Tiree Passage?

Added note in text clarifying particle origin diagnostic, as suggested (p7 line 242). We found that plotting average particle salinity did not provide additional insight to the existing particle origin percentage and particle age figures. The repeated particle releases over the 50-day period resulted in an approximate 50-day mean salinity over the region swept by the particles, but with unpredictable temporal aliasing. Close to the coast, salinity was also very dependent on the interaction between shelf currents and freshwater runoff, which may not be accurately resolved by the model. We considered that a more robust measure of model salinity was to report the arithmetic mean salinity of particles within the observation regions for August and December instead. This was added to the text (p10 line 347).

>10. p.7, line 240: Regarding the statement "While a minority of 20 m particles originated in the abyssal ocean", do you mean that these particles upwelled from off the shelf? The abyssal ocean seems an exaggeration in this context

We intended to convey that the particles originated in deep water and crossed on-shelf, but without vertical displacement. We agree that 'abyssal ocean' infers they originated at depth so have modified the text (p7 line 250).

>11. p.7, lines 257-258: Regarding "the rapid (1-2 day) increase in currents in terms of the dynamic response to wind-induced pressure gradients", I assume you mean geostrophic flows supported by a change in sea surface slope?

Yes, modified text to clarify (p8 line 268).

>12. p.8, first paragraph: Continuing this theme, you suggest rapid setup on a short barotropic adjustment time, perhaps disrupted by inertial effects, but what about Ekman dynamics? Might one expect a strong westerly wind to drive Ekman drift towards Northern Ireland, setting up local downwelling and a geostrophic jet to the west, in the

same sense as the AIC (or ICC in Fig. 1a)? Does this complement on-shelf oceanic flows?

We would expect Ekman drift to result in convergence at the Northern Irish coast which in stratified conditions would drive downwelling and depressed isopycnals. One would associate this density structure with a geostrophic jet to the east rather than to the west in the northern hemisphere? Such a density structure was observed in the AIC core by Porter et al., (2018) in July 2013 so we surmise that this process complements the on-shelf flow as suggested. Text added to acknowledge this feature (p8 line 284).

>13. p.9, lines 302-304: Regarding "between January and March coastal waters are cooler (6-8 _C) than the adjacent ocean (9-10 _C) so we would expect a similar event during this period to increase coastal water temperatures in western Scotland", is there evidence for this in the TPM temperature record?

We do see evidence of this effect, or example in winter 2008. See also figure supplied in response to R2 query, RE impact of storms on TPM long temperature time series. The warming associated with storms is seasonally dependent and more nuanced than salinity. Late season HSPs are sometimes associated with up to 0.5 °C of warming in the TPM time series. A note added to reflect this (p9 line 319).

References

Porter, M., Dale, A. C., Jones, S., Siemering, B. and Inall, M. E.: Cross-slope flow in the Atlantic Inflow Current driven by the on-shelf deflection of a slope current, Deep Sea Res. Part I Oceanogr. Res. Pap., 140, 173–185, doi:10.1016/J.DSR.2018.09.002, 2018.
* * *
[Figure]

FOAM AMM 15
Release date: 12:00 10/01/2019 (100 particles)
Tracked for 20 days

**Fig. 1.** Test particle releases into daily mean (left) and hourly (right) AMM15 model flow fields at 20 m depth.

---

## Author Comment (AC2) · 5 Feb 2020

>L101-104: What is the weight we can attach to two drifter observations? The region has been extensively studied using drifter deployments. Are there other historic records that provide context to the occurrence of such a transport pathway? Further in the MS, the authors use a numerical model to provide context, but I wonder if the historic observational record should also be further explored.

To our knowledge, the relevant historical drifter studies in the region are: i) Burrows and Thorpe (1999) who deployed 42 drifters drogued at 50 m, of which only two crossed onto the Malin Shelf north of Ireland, ii) Booth (1988) in which drifters were entrained in Rockall Trough eddies and did not cross the shelf edge, and iii) Pingree et al., (1999)

[Figure]

who released an Argos float drogued at 45 m in the Celtic Sea which subsequently travelled ~1600 km in the slope current but did not provide any insight into on-shelf flow in the Malin region. While many near-surface drifter trajectories exist, we were primarily interested in deeper flows. We therefore felt that while these studies add important context to understanding of currents in the region, we were justified in considering the present study in isolation. Added references in text to acknowledge the contributions of Booth (1988) and Pingree et al., (1999) (line 68).

>L159-161 & Figure 3: Are these the blue points in Figure 3a? Please edit the caption of Figure 3a appropriately.

Yes, the blue points are particle release locations. Added to label of Figure 3.

>L161-162: Why was the experiment repeated five times? Is this to provide more tracks with a CPU-manageable method, or did the five different experiments have different parameters?

We re-ran the experiment several times to sample a range of diffusive random walks resulting from the diffusive component of particle motion. Clarification added in text (line 164). Also corrected mistake in text as particles were released for 40 days preceding observation period AND during the 10-day observation period. Thus (40+10 days) x 5 repeats = 250 particles released at each location.

>L169: Is T in Equation 2 the same as t in Equation 1?

Yes, modified in text for clarity.

>L170-172: How sensitive are these results to the chosen horizontal eddy diffusivity?

We re-ran the experiment with horizontal eddy diffusivity values between 0.5 and 3 m2 s-1 and found that the results were robust with respect to this coefficient (line 176).

>L202-205: Where does off-shelf transport occur? Does this transport contribute fully to the pathway which leads into the North Sea?

Largely towards the North Sea but the bottleneck imposed by the Minch will force much of the flow around the Outer Hebrides, bringing it back onto the outer shelf. An average off-shelf transport occurs near the base of the water column into the Faroe-Shetland Channel north of 58 N (Figure 2 in Graham et al., 2018). We would surmise that a portion of the imported water makes its way back off-shelf via this region of shelf edge with the rest continuing into the North Sea. We have added a paragraph summarising the above in the manuscript (line 322).

>L212-218 and Figure 9: Although not yet part of the UK Met Office/Met Eirean storm naming, these winter storms were named. I would recommend including their name as part of the text and Figure 9, as these are often used in other analyses: Xaver (5-6 December), Bernd (18-19 December) and Dirk (23-24 December).

Many thanks for this information, storm names have been added to the text and figure captions.

>L226-230: The choice of words here ("backward particle tracking experiments") is confusing. From my understanding, the authors performed a particle tracking experiment where particles were released from a source, and only those tracks which reached the observation location during the observation period were further analysed (see L162- 163).

We agree that this term is confusing and have removed it from the text.

>L244-246: Why were two different experiments used? Are these two from the five mentioned on L161-162? Or are these two times an experiment with 5 particle tracking simulations?

The figures were produced using separate batches of particle release experiments for computational reasons. Each batch consisted of the full release schedule (5 x 50 days = 250 releases per location). Clarified in test (line 254)

> L302-304: Does the TPM record show such occurrences, i.e. where the temperature

also changes in line with the salinity change?

We see evidence that late season HSPs are sometimes associated with up to 0.5 °C of warming in the TPM time series, where temperature and salinity observations overlap. A note added to reflect this (line 319).

> Figure 2 shows gaps in the salinity record at TPM, are more temperature data available which would then potentially allow an analysis of HSP-like events in the temperature record? On L413-414 there is a suggestion that the TPM had temperature data prior to 1994, which could be analysed in such a manner.

The gaps in the salinity record reflect gaps in mooring deployments so impact all observations. The longer temperature time series obtained from current meters at the TPM shows that while the storm events sometimes impact coastal water temperatures, the effect is less pronounced than for salinity. Fig. 1 shows daily temperature anomalies with instances of storm events overlaid. As noted by Inall et al., (2009), the dominant mode of variability in TPM temperature anomalies is inter-annual and is correlated with the upper waters of the Rockall Trough. We did not feel that this figure contributed much beyond the findings of Inall et al., (2009).

>L346-347, Figure 2 and L253: If a cluster of low-pressure systems is a pre-requisite for the occurrence, are HSPs a winter-only phenomenon? How many HSPs occur outside the winter season? Could Figure 2 be edited so the identified HSPs are also plotted on (maybe a marker along the 33.5 or 35.5 salinity level)? I would also recommend changing L253 to say "winter storms".

Based on the available TPM salinity data, HSPs are a winter-only phenomenon. This is probably due to reduced frequency of storms in the summer, coupled with an increase in inner-shelf stratification which presents more of a barrier to the ingress of oceanic water. Note added in text (line 397). Ticks also added to Fig. 2 to highlight HSPs as suggested.

[Figure]

>Bathymetry contours in figures: The model bathymetry is based on the EmodNet data product. I would suggest contours based on this product in all plots.

Agreed; plots depicting model data recreated with EmodNet bathymetry.

>Figure 2: In the discussions PDF, this figure didn't occupy the full width. This could be due to the editorial system, but I would recommend for the authors to check there is no unnecessary white space in the image. I think this figure merits a full A4 width space, to make sure it is legible.

Modified as recommended.

>Figure 8, caption: please add "(in red)" after "Bathymetry contours". As far as I could tell, this was also only the 500 m one, so I would suggest "Bathymetry contour (red) ... "

Modified as recommended.

References

Booth, D.A., 1988. Eddies in the Rockall Trough. Oceanologica Acta, 11(3), pp.213-219. Burrows, M. and Thorpe, S. A.: Drifter observations of the Hebrides slope current and nearby circulation patterns, Ann. Geophys., 17(2), 280–302, 1999.

Graham, J. A., Rosser, J. P., O'Dea, E. and Hewitt, H. T.: Resolving Shelf Break Exchange Around the European Northwest Shelf, Geophys. Res. Lett., 45(22), 12,386-12,395, doi:10.1029/2018GL079399, 2018.

Inall, M., Gillibrand, P., Griffiths, C., MacDougal, N. and Blackwell, K., 2009. On the oceanographic variability of the North-West European Shelf to the West of Scotland. Journal of Marine Systems, 77(3), pp.210-226.

Pingree, R.D., Sinha, B. and Griffiths, C.R., 1999. Seasonality of the European slope current (Goban Spur) and ocean margin exchange. Continental Shelf Research, 19(7), pp.929-975.

Porter, M., Dale, A. C., Jones, S., Siemering, B. and Inall, M. E.: Cross-slope flow in the Atlantic Inflow Current driven by the on-shelf deflection of a slope current, Deep Sea Res. Part I Oceanogr. Res. Pap., 140, 173–185, doi:10.1016/J.DSR.2018.09.002, 2018.

[Figure]

[Figure]

Tiree Passage Mooring (TPM) seasonal temperature anomaly
(calculated using day of year). Grey bars denote DJFM, green
bars show storm events (daily mean westerly wind > 18 m s-1)

**Fig. 1.** Temperature anomalies at the Tiree Passage Mooring, at 20 m. Grey bars denote winter months (DJFM). Green lines show instances of daily mean westerly winds on Malin shelf exceeding 18 m s-1.